# LART: Neural Correspondence Learning with Latent Regularization Transformer for 3D Motion Transfer

**Haoyu Chen[1]    Hao Tang[2,3]    Radu Timofte[2,4]    Luc Van Gool[2,5,6]    Guoying Zhao[1]***

[1]CMVS, University of Oulu    [2]Computer Vision Laboratory, ETH Zurich
[3]Carnegie Mellon University    [4]University of Wurzburg    [5]ESAT-PSI, KU Leuven
[6]INSAIT, Sofia Un. St. Kliment Ohridski

chen.haoyu@oulu.fi, hao.tang@vision.ee.ethz.ch, radu.timofte@uni-wuerzburg.de
vangool@vision.ee.ethz.ch, guoying.zhao@oulu.fi

## Abstract

3D motion transfer aims at transferring the motion from a dynamic input sequence to a static 3D object and outputs an identical motion of the target with high-fidelity and realistic visual effects. In this work, we propose a novel 3D Transformer framework called LART for 3D motion transfer. With carefully-designed architectures, LART is able to implicitly learn the correspondence via a flexible geometry perception. Thus, unlike other existing methods, LART does not require any key point annotations or pre-defined correspondence between the motion source and target meshes and can also handle large-size full-detailed unseen 3D targets. Besides, we introduce a novel latent metric regularization on the Transformer for better motion generation. Our rationale lies in the observation that the decoded motions can be approximately expressed as linearly geometric distortion at the frame level. The metric preservation of motions could be translated to the formation of linear paths in the underlying latent space as a rigorous constraint to control the synthetic motions occurring in the construction of the latent space. The proposed LART shows a high learning efficiency with the need for a few samples from the AMASS dataset to generate motions with plausible visual effects. The experimental results verify the potential of our generative model in applications of motion transfer, content generation, temporal interpolation, and motion denoising. The code is made available: https://github.com/mikecheninoulu/LART.

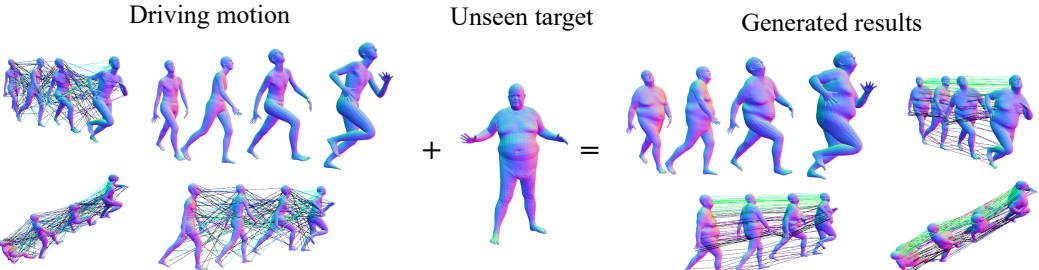

Figure 1: Given as driving mesh sequences with random-ordered vertices, our learned LART could transfer the motion from the driving sequences to the unseen 3D targets without any further fine-tuning. Here we connect the points based on the vertex order to verify that our LART does not need the pre-defined correspondence, which is vertex-order invariant. Samples are from the AMASS dataset [21].

---

*Corresponding author.

37th Conference on Neural Information Processing Systems (NeurIPS 2023).

# 1 Introduction

3D motion transfer is one of the most challenging 3D generation tasks. It refers to the task of transferring desired motions from a driving sequence to a static 3D subject and outputting the corresponding high-fidelity and realistic synthetic dynamics. Aside from the research interests, it has countless applications across areas of interest, i.e., 3D animated character design in the film industry, VR/AR enhancement, emotion transfer, or real-time remote interaction [38]. It is desirable that one model can automatically generate realistic animated 3D avatars without manually specifying the geometry correspondences and dynamics, which would be cumbersome when using a standard graphics rendering engine.

Although existing works can achieve 3D motion transfer via various fashions with plausible and promising visual results, the question of automatic 3D dynamic generation is far from being closed. There are several major limitations among existing methods. First, the correspondence between source and target meshes needs to be provided, such as point-wise correspondence [32], human key point annotations [2], or template registrations [36] (e.g., SMPL [4]). Unfortunately, these all involve costly manual data annotations and are not always available. Second, most of the existing models strongly rely on the 3D shape prior learned from the training sets to keep the intrinsic attributes [13, 40, 28, 8], and cannot generalize to unseen targets that are not in the training set. Thus, collecting new samples each time is a must to train a model for each new subject or shape style many [23]. This would make the model not easily scalable. At last, the generation of surfaces needs fine geometric details to be preserved while transferring the extrinsic motions through a highly-nonlinear map.

We cast the task of 3D motion transfer as a continuous deformation problem, where the goal is to train a model such that the target 3D object has identical dynamics to driving sequences while preserving its appearance information (see Figure 1). In particular, we design a new 3D Transformer structure with cross-attention modules to continuously learn the implicit correspondence between target meshes and dynamics from the driving sequences. It also has an adaptive feature encoder with an efficient positional embedding scheme so that it can handle large-size meshes with fine geometric details fully preserved. Another key contribution is a novel latent metric regularization on the Transformer. Inspired by recent geometric deep learning methods [11, 14] that inject geometric regularization in the training, we formulate the decoded motions in the temporal domain as an approximation of linearly geometric distortion at the spatial domain on isometric pairs (i.e., just a change in pose). By coupling the Euclidean distances among latent codes (hence, along linear paths in the latent space) to the metric distortion among decoded motions, we obtain a strong regularizing effect in the construction of the latent space and, in turn, significantly reduce the need for large training datasets. Our method can learn to generate high-fidelity, realistic, and temporally coherent 3D human motions. Moreover, we show it is possible to animate completely unseen 3D mesh with randomly ordered-vertex motion sequences, which is extremely challenging. Lastly, we extend our method to other domains, such as animals and different tasks.

In summary, our main contributions are:

- We work on the 3D motion transfer problem, especially for unseen target meshes. To the best of our knowledge, our method is the first Transformer-based 3D motion transfer framework that learns the correspondence implicitly between meshes.
- We propose the LART, a novel Transformer-based architecture for 3D motion transfer, carefully designed to generate fully-preserved geometric details. A novel feature encoder with an adaptive positional coding scheme is introduced to handle large-size meshes and reduce the dimensionality redundancy and computation cost. It can also implicitly learn the correspondence between meshes via an across-attention layer.
- A novel latent geometric regularization on the Transformer is presented for synthesizing realistic dynamic results. It formulates the temporal generation problem as continuous deformations by coupling the Euclidean distances among latent codes to the metric distortion among decoded motions as a tight constraint, allowing models to gain generability efficiently and process driving sequences with flexible lengths.
- Quantitative or qualitative experimental results on four different datasets show that the proposed method achieves satisfying performances with substantially fewer training samples. Moreover, we further represent the capability of generalizing our framework to other domains and various generative applications, showing it a strong baseline for data-driven 3D motion transfer.

## 2 Related Work

**3D Mesh Deformation** is closely related to our task. It aims to generate a new 3D shape with a given pair of source and target shapes, which can be seen as a frame-level task of 3D motion transfer. Towards 3D mesh deformation, previous works demand re-enforcing the correspondence between source and target meshes. For example, some disentanglement-based methods [11, 40] use the shape correspondences between different pose meshes from the same body to decompose shape and pose factors. Furthermore, Gao et al. [13] proposed using cycle consistency to achieve the pose transfer. Aigerman et al. [1] introduced the neural Jacobian field with mesh-specific, basic linear-algebra operations to an intrinsic field of matrices to preserve the detailed geometry, but it relies heavily on discrete differential geometry operators, which need rigorous processed watertight meshes. Generally, all the above methods cannot deal with unseen identities due to the strong dependence on the shape priors. Differently, recent methods [35, 9] utilize the latest technique for image style transfer to achieve the neural pose transfer without the need for other guidance. 3D-CoreNet [31] explicitly learns the correspondence and refines the generated meshes jointly by solving the optimal transporting problem. However, it will be computationally costly when processing multiple continuous meshes for the motion transfer. In contrast, we use a structured adaptive encoder to learn the correspondence implicitly. Most importantly, our model involves the temporal consistency from the sequential meshes, which is challenging and not touched by those 3D deformation methods.

**Generalized 3D Animation.** Generating 3D mesh sequences via motion information has been intensively researched from different aspects. One main direction closely related to our task is to use the linear blend skinning (LBS) method [4] and the recent implicit skinning networks to construct animatable models. For instance, SNARF [10] uses differentiable forward skinning for animating non-rigid neural implicit shapes, which can be extended to unseen and complex shapes. SCANimate [28] learns a pose-aware parametric clothed human model by introducing a cycle procedure to achieve the learning of LBS. Similarly, MetaAvatar [36] can learn an animatable model with only a few depth images. However, animation of those methods shares a strong prerequisite that certain registrations, such as bone transformations between the target meshes, are inevitable. The motion is achieved by manipulating SMPL pose parameters or key joints of the learned parametric human body models. Thus inherently cannot transfer motion from raw driving mesh sequence to unseen targets. Differently, our learned model can directly conduct motion transfer to unseen 3D meshes from different datasets without any further fune-tine, which is much more challenging.

**3D Transformer with Temporal Modeling.** Transformer architecture was first proposed by [34] and raised emerging research interests in vision tasks with promising performances [16, 12]. To retain positional information of sequential dependencies, a trainable linear projection layer to embed each input patch to a high dimensional feature is always applied, such as pose estimation [39], motion synthesis [25]. However, we argue that the 3D animation task differs from the above tasks. 3D surfaces, as opposed to the standard 2D pixel grid, have significant geometric and topological variations. Standard 2D convolutional architectures make assumptions about point ordering and parametrization that do not apply, in general, to collections of 3D surfaces. Thus, it's crucial that the network architecture can adaptively perceive the geometry and topology. Furthermore, the generation of the 3D surfaces needs detailed preservation. Some 3D motion synthesis methods, such as ACTOR [25] and Action2Motion [15], learn generators based on VAE [20] that achieve the action-conditioned generation of realistic and diverse 3D human mesh sequences. Again, an SMPL joint regressor is introduced to ACTOR to ensure the realistic generated motions, and Action2Motion post-renders the skeleton joints to SMPL models to obtain better-visualized results. Apparently, those methods fail to capture complex and geometric details, such as the wrinkle of the human cloth models. Thus, we propose a novel Transformer architecture, the LART, for the 3D motion transfer task. Compared to the other works, we have two core differences: 1) both the appearance details of target meshes and desired motions can be handled in the generated 3D sequences, and 2) our framework directly processes the raw mesh sequences without any manual intervention (e.g., SMPL parameters or skeleton joints) in an end-to-end manner.

## 3 Data-Driven 3D Motion Transfer

Inspired by previous 2D video synthesis works [37, 29] for animating still 2D images and 3D pose transfer [35], we are interested in automatically transferring the motion of a driving sequence

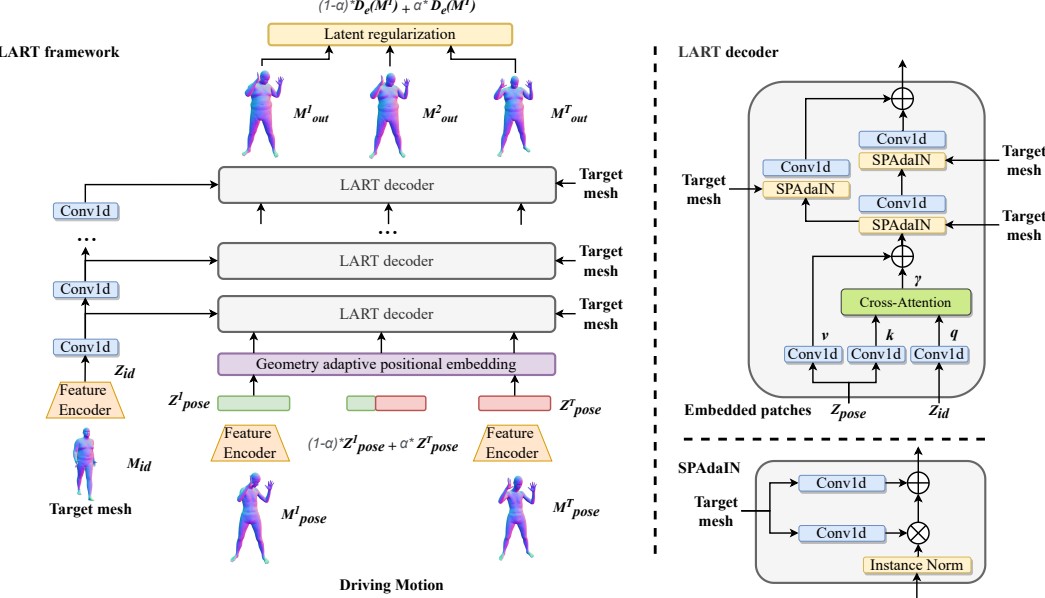

Figure 2: Architecture overview of LART. Left part: the whole LART framework. We process the sequential driving meshes into features, add geometric adaptive positional embeddings, and feed the resulting sequence of vectors to our LART decoder. The resulting generated meshes will be regulated via latent metrics with ground truths. Right part: the detailed architectures of an LART decoder and a SPAdaIN block.

to a static 3D target. We first present a general introduction to the LART framework for neural correspondence learning. Then, we will demonstrate how to impose the latent regularization on the LART for training.

## 3.1 LART: Latent Regularization based Transformer with Neural Correspondence Learning

In the task of 3D motion transfer, especially for an unseen target setting where the shape prior is not available to build a compact representation of the latent space, we argue that the latent representations of 3D models should be sufficient and strong enough to preserve and manipulate the detailed surface geometry of the target mesh. Inspired by NPT and NeuralBody [35, 24], the idea of using conditional normalization directly in the spatial sense and preserving the structured topology latent codes for detailed geometry capturing is very intuitive and proven effective. However, this will make the latent presentation of the meshes considerably big and make it difficult to model the temporal information, as the number of learnable parameters of conventional recurrent temporal models like RNNs/LSTM [18] grow quadratically with the memory size, making them extremely large and almost non-trainable.

Recently, the Transformer architectures [12] have become popular as they allow parallel computation inputs, which both speed up the training and save the memory, especially suiting the needs of our task. Thus, we chose a Transformer-based architecture to build up our model, which consists of three main components, a geometry adaptive feature encoder, a LART decoder, and a latent metric regularization. An overview of the LART is depicted in Figure 2.

**Geometry Adaptive 3D Feature Encoder.** Given a 3D mesh driving sequence of $T$ frames $M_{pose}^1, \cdots, M_{pose}^T$, where $M_{pose} \in \mathbb{R}^{N \times 3}$ and $N$ stands for the vertex number, we use the same 3D feature extractor as previous works [35, 31, 9] to extract a static 3D feature per frame. The feature encoder consists of three blocks of 1D convolution layers followed by Instance Norm, resulting in a sequence of latent embedding vector $Z_{pose}^1, \cdots, Z_{pose}^T$, where $Z \in \mathbb{R}^{C \times N}$ and $C$ is the channel dimension (e.g., 1,024). The network weights of the feature encoder are shared for all frames. At the same time, we obtain the latent code $Z_{id}$ of target mesh $M_{id}$ via the same feature encoder. Next, we will embed the features of the driving motion at each frame with our geometry adaptive positional embedding scheme as below.

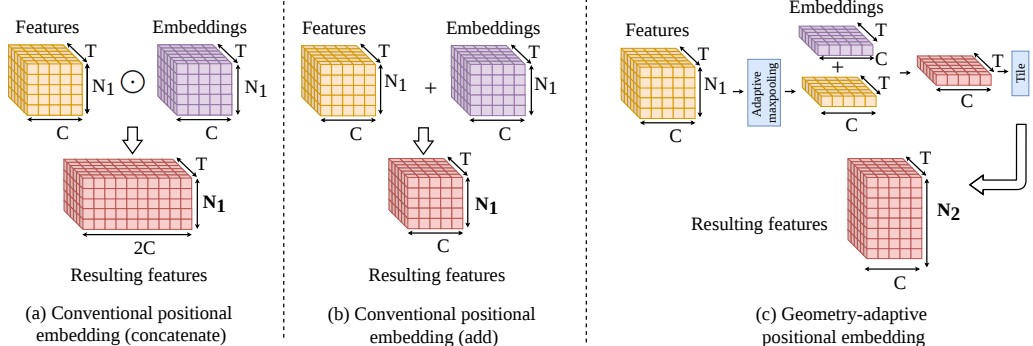

(a) Conventional positional embedding (concatenate)

(b) Conventional positional embedding (add)

(c) Geometry-adaptive positional embedding

Figure 3: A comparison of different positional embedding methods. Note that our methods can output a flexible size of embedded features so that it can be extended to unseen targets with different vertex numbers than the training set.

As shown in Figure 3, to retain the positional information of the sequence, there are commonly two main ways ("concatenating" and "adding"), and "adding" one is more popular as it significantly saves memory [17, 39]. However, 3D motion transfer needs to maintain various sizes of the target meshes (vertex number as $N_2$) with motion sequential meshes (vertex number as $N_1$, $N_1 \neq N_2$), leading to the need for a flexible coding scheme. These two embedding ways with fixed shapes will hinder the model from extending to different vertex sizes and are apparently not feasible in our case. Thus, we proposed a simple but efficient positional embedding scheme for maintaining the features of structured topology. To ensure the network's generalizability to different sizes of unseen meshes while preserving the temporal information, we first conduct adaptive max pooling to every pose feature $Z_{pose}^t$ (to the mesh at $t$ frame of motion) to obtain a global-wise latent pose code, and then add the same size vector to it as positional/temporal embedding and expending the feature along the topology dimension ($N_2$) as the target mesh has. In this way, we obtain a new feature vector of the pose that has both positions embedded and the same size as the target mesh's, making the learning of the correspondence between two meshes with different sizes possible ($N_1$ and $N_2$). After the positional embedding, the resulting sequential feature vectors $Z_{pose}$ together with a single feature vector $Z_{id}$ is sent to the LART decoder.

**LART Decoder.** As discussed in the work of 3D-CoreNet [31], it's beneficial to learn the correspondence between the given target mesh and driving mesh sequences. However, 3D-CoreNet needs to explicitly learn the correspondence between the meshes by solving the optimal transporting problem. This could be redundant as we need to empirically set hyper-parameters, i.e., iterations in the Sinkhorn algorithm [30]. Instead, we improve the attention mechanism of the Transformer to implicitly learn the implicit correspondence, thus, without the need for extra intervention. Specifically, as shown in Figure 2 right part, we feed the feature vectors from the target and pose meshes into two different 1D convolution layers to generate the representations $\mathbf{q}, \mathbf{k}, \mathbf{v}$ as that in the standard self-attention [34]. The query $\mathbf{q}$ is from $Z_{id}$, and the value $\mathbf{v}$ and key $\mathbf{k}$ are from $Z_{pose}$. Then, the attention weights $A_{i,j}$ based on the geometric pairwise similarity between two elements of $\mathbf{q}$ and $\mathbf{k}$ is given with the following formula:

$$\mathbf{A}_{i,j} = \frac{exp(\mathbf{q}_i \mathbf{k}_j)}{\sum_{i=1}^{n} exp(\mathbf{q}_i \mathbf{k}_j)}. \tag{1}$$

After this, a matrix multiplication between $v$ and the transpose of $\mathbf{A}$ is conducted to perceive the geometric inconsistency between meshes. Finally, we weigh the result with a scale parameter $\gamma$ and conduct an element-wise sum operation with the original latent embedding $Z_{pose}$ to obtain the refined latent embedding $Z'_{pose}$,

$$Z'_{pose} = \gamma \sum_{i=1}^{n} (\mathbf{A}_{i,j} \mathbf{v}_i) + Z_{pose}, \tag{2}$$

where $\gamma$ is initialized as 0 and updated gradually during the training with gradients. The obtained $Z'_{pose}$ is followed by typical Transformer structures as introduced above Figure 2 with a convolutional layer and SPAdaIN block [19, 35]. Though multiple stacked LART decoders with latent code size gradually decrease, we generate a sequence of final output $M_{out}^t$.

In such a crossing way, the geometric-perceived feature code can consistently be rectified by the original identity mesh and its latent embedding representations. Note that input mesh vertices are

all shuffled randomly, while our network can output vertex-aligned meshes, showing that LART is vertex-order invariant and can successfully learn the correspondence between meshes, see Figure 1. Please refer to the Supplementary Materials for more implementation details.

## 3.2 Latent Metric Regularization

Here, the efforts in this section are motivated by the observation that 3D motion transfer is different from 3D pose transfer or 3D deformation. There is a lot of redundant information in the motion. Thus, directly implementing existing data-driven approaches for 3D motion transfer, completely relying on the training dataset, will impose a heavy burden on the learning process. For instance, the geometric distortion within a motion clip of 30 frames might be considered subtle, making it extremely inefficient to construct the latent pose space by traveling all the training motions. On the other hand, the meshes within the same motion are isometric (i.e., just a change in pose) and share the same intrinsic information, which naturally could be utilized as a regularization in training.

With the above observations, we adopt the idea of spatial deformation regularization from LIMP [11] into a temporal case. Specifically, we regard the presence of a motion as a continuous interpretation problem of the pose at each frame, which can be regularized by metric interpolation of the pose space.

**Treating Motion as Continuous Interpretation.** In the continuous setting, we give the latent metric regularization as follows:

$$\mathcal{L}_{Metric} = \|\mathbf{D}_e(dec((1-\alpha)Z^1_{pose} + \alpha Z^T_{pose}) - ((1-\alpha)\mathbf{D}_e(M^1) + \alpha \mathbf{D}_e(M^T)))\|^2_2, \quad (3)$$

where $\alpha \sim \mathcal{U}(0,1)$ is a uniformly sampled scalar in (0, 1), standing for the index frame of a motion. The $dec$ operator is to decode the latent code $Z$ into mesh $M_{out}$ via the LART decoders. In the equation above, the matrix $\mathbf{D}_e$ encodes the pairwise Euclidean distances [26].

At training time, given two poses $M^1$ and $M^T$ from a motion. We then assume there exists an abstract metric space where each point is a pose; this "pose space" is the latent space that our generative model seeks to learn. By constructing the latent code with a parametric sequence of poses $Z^\alpha_{pose} = (1-\alpha)Z^1_{pose} + \alpha Z^T_{pose}$, connecting $M^1_{out}$ to $M^T_{out}$, we regard each $Z^\alpha_{pose}$ as a continuously deformed version between $M^1_{out}$ to $M^T_{out}$. By using the metric loss of Eq. (3), we can enforce the latent code and Euclidean distance of the motion to be linearly consistent. As a result, as $\alpha$ grows from 0 to 1, we are modeling a sequence of approximate isometries as a motion.

**Manipulating on the Flattening Latent Space**. Like previous works [11, 33], our learning model performs a "flattening" operation, imposing the latent space to be as Euclidean as possible. The expression of the motion can be linearly manipulated by varying the $\alpha$. This enables algebraic manipulation of the latent codes for various applications, such as motion interpolation and denoising.

## 3.3 Full Objective Function

With the latent metric regularization loss above, we define the full objective function as below:

$$\mathcal{L}_{full} = \lambda_{rec}\mathcal{L}_{rec} + \lambda_{edge}\mathcal{L}_{edge} + \lambda_{metric}\mathcal{L}_{metric}, \quad (4)$$

where $\mathcal{L}_{rec}$, $\mathcal{L}_{edge}$ and $\mathcal{L}_{metric}$ are the three losses used as our full optimization objective, including reconstruction loss, edge loss, and our newly proposed latent metric regularization loss. $\lambda$ is the corresponding weight of each loss. In Eq. (4), reconstruction loss $\mathcal{L}_{rec}$ is the point-wise L2 distance and the edge loss [14] is an edge-wise regularization between the GT meshes and predicted meshes. The mathematical definitions can be found in the Supplementary Materials.

## 4 Experiments

In this section, we perform an extensive evaluation of the proposed LART. Quantitative and qualitative experimental results on four different datasets show that the proposed method achieves satisfying performances with substantially fewer training samples. Moreover, we further represent the capability of generalizing our framework to other domains, such as animals and various generative applications.

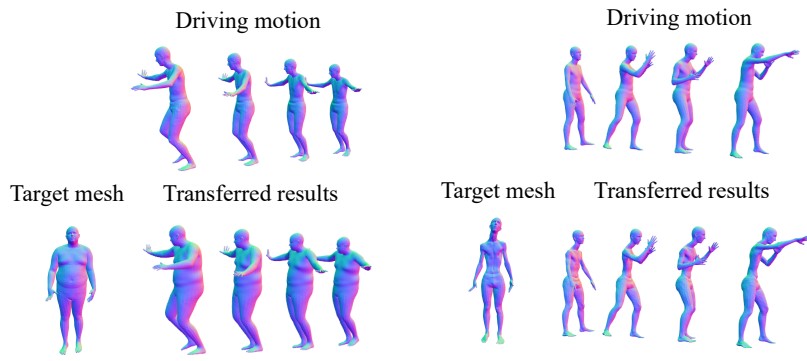

**Motion transfer to seen target**     **Motion transfer to unseen target**

Figure 4: Qualitative results of 3D motion transfer by LART. The left part shows the motion transfer to seen target in training, and the right part shows the motion transfer results to the target mesh, which is unseen in the training set.

Table 1: Comparison with the state-of-the-art methods on the DFAUST datasets. We compute frame-level average point-wise mesh Euclidean distance (PMD) as the metric. Our LART outperforms other compared methods by a large margin with more efficient learning on the training sets.

| Method | PMD↓ ($\times 10^{-4}$) | |
|---|---|---|
| | Seen Motions | Unseen Motions |
| NPT-MP [35] | 4.51 | 6.69 |
| NPT [35] | 4.37 | 5.31 |
| 3D-CoreNet [31] | 1.25 | 3.05 |
| LART (Ours) | **0.41** | **1.80** |

## 4.1 Datasets

**Driving Motion Datasets.** For training the model, we use the motion sequences from the DFAUST [6] and AMASS [21] datasets. DFAUST is a 3D human motion dataset with 40,000 raw and aligned meshes, captured at 60 fps. There are ten human subjects with 129 different body motions, such as "punching", "chicken wings pose", etc. Each motion consists of hundreds of frames. AMASS is a large and varied database of human motion that contains 15 different sub-datasets (including SMPL-registered DFAUST), spanning over 300 subjects with more than 1,1000 motions. Here we use the DFAUST dataset as a quantitative evaluation and select some representative motions from AMASS for training and presenting the qualitative results. To create training data, we randomly pair one motion (driving sequence) and one shape (target mesh) for training. The corresponding ground truth is obtained by using the SMPL model [4] to synthesize the target animated sequence with the shape and pose parameters for evaluation.

**Target Datasets.** In testing, we use the learned LART model to transfer motions from given driving sequences to unseen target meshes. To evaluate the generalization ability of our LART, we employ the learned model to drive the target meshes from other unseen datasets, e.g., FAUST [5] and MG-dataset [3], for qualitative evaluation. For quantitative evaluation, we use eight new target meshes generated with the SMPL model by sampling from pose and shape parameter spaces. Then we conduct the 3D motion transfer on these new target meshes. We evaluate the model with two protocols on the DFAUST, i.e., the seen driving sequences that appear in the training set and the unseen driving sequences that are not in the training set.

**Other Domains.** Lastly, we also use LART to achieve motion transfer on different domains other than human meshes, such as animal meshes from the Animal dataset [32] and hand meshes from the MANO dataset [27] via domain-specific training. The mesh vertices are all shuffled randomly to ensure the network is vertex-order invariant. The training is achieved in a supervised manner, while in the testing, we directly conduct the motion transfer to unseen meshes with the trained model. See more details on how we process the datasets in the Supplementary Materials.

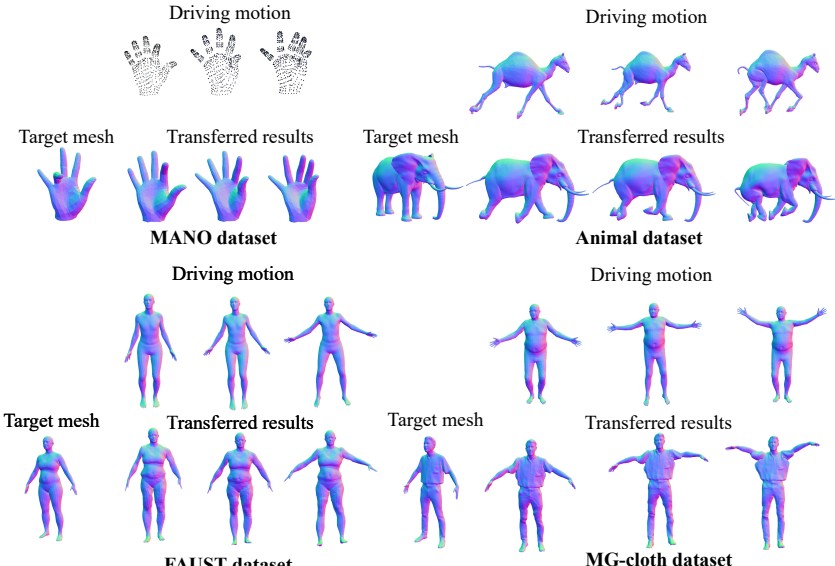

Figure 5: Generalization ability of LART. On the top, we show the domain generalization ability of LART by conducting 3D motion transfer on 3D hands from MANO dataset [27], and 3D animals from Animal dataset [32]. On the bottom, we show the generalizing ability of LART to unseen mesh targets from the FAUST dataset and MG-cloth dataset. Note that meshes in MG-cloth have more than 27,000 vertices with fine-grained geometric details, showing that LART works without the need to comply with a typical SMPL template which has 6,890 vertices.

## 4.2 Quantitative Evaluation

**Evaluation Metrics.** We chose the commonly used point-wise mesh Euclidean distance (PMD) as the evaluation metrics [35, 9]. It is the L2 distance between the vertices of the output mesh and the ground truth mesh. In the 3D motion transfer task, we obtain the average PMD over all the frames:

$$PMD = \frac{1}{T}\frac{1}{N}\sum_{\mathbf{t}}\sum_{\mathbf{n}}\left\|M_{\mathbf{n}}^t - G_{\mathbf{n}}^t\right\|_2^2, \qquad (5)$$

where $M_{\mathbf{n}}$ and $G_{\mathbf{n}}$ are the point pairs from the generated mesh $M$ and ground truth mesh $G$.

**Baselines.** We compare the proposed approach to the most recent single mesh deformation approaches. Specifically, we compare three state-of-the-art methods, i.e., NPT [35], 3D-CoreNet [31]. For these two methods, we use them to generate long sequences frame by frame as they are originally for 3D pose transfer without considering the temporal domain. Note that we use the same protocols provided to train the models to make sure they are optimized to the greatest extent.

**Training and Runtime.** Our network receives a sequence of driving meshes of size $N_1 \times 3 \times T$, and a target mesh of size $N_2 \times 3$ where $N$ and $T$ stand for the vertex number and frame number. Note that $N_1$ might not equal $N_2$ because our LART can handle meshes of different sizes. The network is implemented in PyTorch [22] and optimized using the Adam. The batch size is 2. The total number of training epochs is 200. The learning rate is initialized as 5e-5 and reduced at milestones of 80, 120, and 160 with gamma as 0.1. During the training, we vary the $\alpha$ to better learn the underlying linear path in the latent space. The length of the driving sequences is fixed as three frames for the computational memory issue. In the first phase (100 epochs), LART takes as input all the continuous frames from a motion clip to stabilize the training. Then, in the later phase, we add latent regularization which allows us to skip frames between the start and end of the motion by encoding them linearly. The total training time on the DFAUST dataset is around 31 hours on an Nvidia GPU V100, shorter than other methods [35, 31, 7]. During inference, we fix $\alpha$ as 0.5 for both 3D motion transfer and temporal interpolation tasks. We conduct the long sequential motion transfer to the target mesh via a sliding window. Inference runs at ~120ms per frame on DFAUST (6,890 vertices).

**State-of-the-Art Comparisons.** We provide experimental results by comparing LART with state-of-the-art methods quantitatively in Table 1. For both settings: "Seen motions" and "Unseen motions"

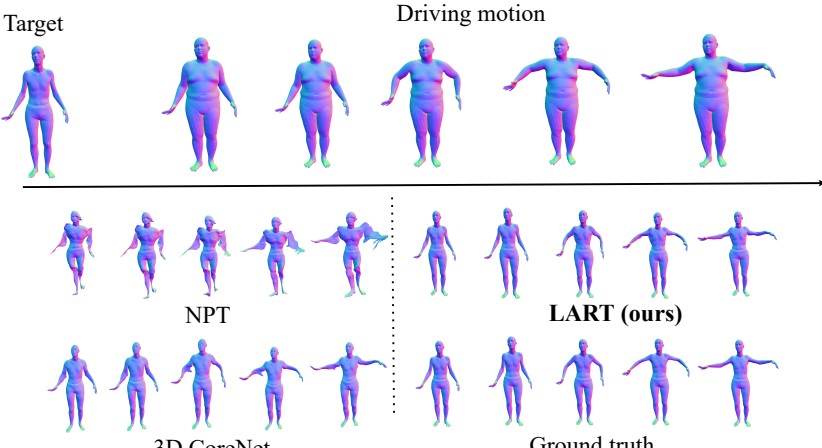

Figure 6: Qualitatively comparison with the state-of-the-art methods on motions from the AMASS dataset. Given the long driving sequence, compared methods process the deformation frame by frame while LART takes several frames as inputs.

Table 2: Ablation study to latent metric regularization (LMR) by verifying the performances of a single subject from the DFAUST dataset.

| Method | NPT | LART single frame | LART w/o LMR | LART (Ours) |
|---|---|---|---|---|
| Training Time | 21 h | 25 h | 20 h | **8 h** |
| PMD↓ ($\times 10^{-4}$) | 2.82 | 2.77 | 1.90 | **0.32** |

from the DFAUST dataset, our LART significantly outperforms compared state-of-the-art methods with frame-level average PMD ($\times 10^{-4}$) of: 0.41 and 1.80 (LART) vs. 1.25 and 3.05 (3D-CoreNet). As shown in the Figure 6, given unseen motion sequences, both NPT and 3D-CoreNet suffer from degeneracy, while LART still can provide relatively robust results.

**Ablation Study.** We provide an ablation study in Table 2. As shown, we can further verify the effectiveness of each component in the LART, with LMR not only improving the model performance but also shortening the training time.

### 4.3 Qualitative Evaluation

**Non-SMPL Mesh Motion Transfer.** LART can be generalized to the animation of non-SMPL meshes. As shown in Figure 5, LART can directly conduct motion transfer to unseen meshes from FAUST [5] and MG-cloth [3] datasets without any further finetuning. Note that those meshes are not in line with the SMPL model and are more challenging (more fine-grained geometry details). Furthermore, we show the results of our model on animating animals and hands in Figure 5, by training on the Animal and MANO datasets, showing the capacity of our LART in different domains.

**Potential Applications.** Furthermore, we show two potential applications of LART in Figure 7. For denoising, we add Gaussian noise to each vertex of the input mesh. Results show that LART can not only handle random ordered points as input but is also insensitive to noise to some extent (even though some artifacts can be observed on the arm part of the human mesh). For the temporal interpolation, LART takes the start and end frames as input; it can complete the middle frames.

**Limitations.** One main issue of LART is the computational complexity. To capture the detailed geometries of an unseen target, we use structured latent codes (geometric adaptive embedding) along with the target mesh to have a better performance. However, it is memory-consuming. Besides, since the setting of LART is to directly transfer motion to unseen targets and generate the animation, LART cannot capture and generate the subject/pose-dependent dynamics such as the plausible wrinkles of the cloth and subtle muscle moments. This kind of non-linear local deformation that can efficiently improve the generative visual effects has not been explored in this work yet, which could be the future

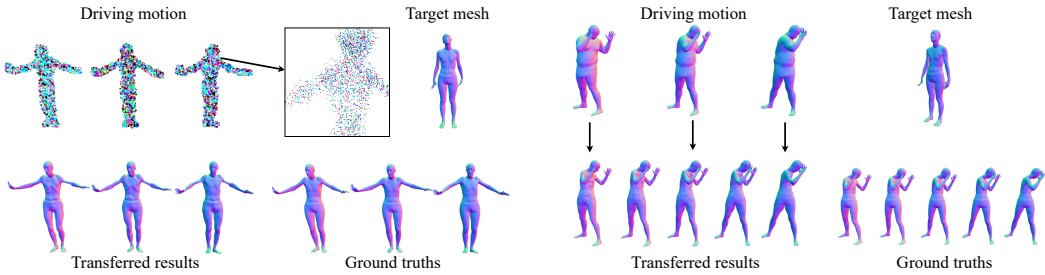

Figure 7: Potential examples of our LART application. The left side shows a denoising use case of LART. Given a noisy driving sequence, LART can still handle and complete the information and output relatively robust results. The right side shows using LART to interpolate an incomplete sequence.

research direction. Another limitation is that LART (and all other baselines) relies on the latent pose space learned to generate high-quality results. If a motion is far away from the learned distribution, it will lead to degenerated results. Although efforts have been made here to boost latent space learning, the problem of efficiently covering the whole latent space is far from being closed.

## 5   Conclusion

In this work, we propose the LART framework, i.e., a novel Transformer network customized for 3D motion transfer with carefully-designed architectures. LART can implicitly learn the correspondence and can also handle large-size full-detailed unseen 3D targets. Besides, we introduce a novel latent metric regularization for better motion generation. Our proposed LART shows a high learning efficiency with promising visual effects. The experimental results verify the potential of our generative model in applications of motion transfer, content generation, temporal interpolation, and motion denoising. The future direction could be compact architecture design and efficient latent space learning as strong priors to unseen domains.

## 6   Acknowledgement.

This work was supported by the Research Council of Finland (former Academy of Finland) Academy Professor project EmotionAI (grants 336116, 345122), ICT 2023 project TrustFace (grant 345948), the University of Oulu & Research Council of Finland Profi 7 (grant 352788), and Infotech Oulu. As well, the authors wish to acknowledge CSC – IT Center for Science, Finland, for computational resources. Support by the Alexander von Humboldt Foundation and by the Bulgaria government is also gratefully acknowledged.

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
