# OpenReview forum: "LART: Neural Correspondence Learning with Latent Regularization Transformer for 3D Motion Transfer"
_NeurIPS.cc/2023/Conference — NeurIPS 2023 poster_

### Official Review · Reviewer_cJpb · 2023-07-02

**Soundness:** 2 fair
**Presentation:** 3 good
**Contribution:** 2 fair
**Rating:** 5
**Confidence:** 3

**Summary:**

1.The paper presents LART, a 3D Transformer framework for 3D motion transfer. One of the distinctions from previous methods is that LART does not require joint annotation or pre-defined correspondence between the source and target mesh. By preserving motion metrics and effectively controlling synthetic motions in the latent space construction, LART achieves accurate motion synthesis.  The experimental results demonstrate the high learning efficiency of LART, requiring only a few samples from the AMASS dataset to generate motions with plausible visual effects.

**Strengths:**

1. The paper is well-written and presents the information in a clear and comprehensible manner.

2. A novel latent geometric regularization is proposed for synthesizing realistic dynamic results.

3. The proposed method demonstrates its versatility by successfully extending its applicability beyond human motion transfer to animal motion transfer, as illustrated in Fig 5.

4. The method achieves good performance in both quantitative and qualitative evaluations, providing solid evidence of its effectiveness.

**Weaknesses:**

1. The author should provide more explanation and necessary information about specific terms and blocks used in the paper. For instance, a detailed description of the SPAdaIN block is needed, including its purpose and relevance in the decoder. Additionally, the author should further highlight that SPAdaIN is from [33], not just cite it without mention.

2. The design of the cross-attention mechanism depicted in Fig 2 is not clearly described. More detailed explanations regarding its implementation are required.

3. The proposed encoder is named "geometry adaptive 3D feature encoder," but it seems that geometry is only involved in the positional embedding. It would be beneficial to consider incorporating a geometry-aware design in the encoder, in addition to the positional embedding.

4. The paper lacks an ablation study on the three different positional embedding methods presented in Fig 3. It is important to analyze the performance of each positional embedding method, as the supplementary Table 7 only reports the loss without providing any analysis.

5. While the paper includes visualizations in figures, it would be advantageous to provide a demo video to showcase the actual quality of the visualizations.

**Questions:**

See weakness

**Limitations:**

See weakness

---

> ### Author Rebuttal · Authors · 2023-08-10
>
> Thank you for your acknowledgment of the novelty of our work and the constructive feedback! We will address your questions and concerns in the following:
>
> **Q1: The author should provide more explanation and necessary information about specific terms and blocks used in the paper. For instance, a detailed description of the SPAdaIN block is needed, including its purpose and relevance in the decoder. Additionally, the author should further highlight that SPAdaIN is from [33], not just cite it without mention.**
>
> A1: We appreciate your valuable suggestion that can help us to illustrate our work for the following researchers better. SPAdaIN was first proposed by Wang et al. in [33] by injecting the original target mesh into each layer of the network with a residual-like connection. In this way, the detailed spatial geometry can be perceived throughout the whole flow of the network. Thus, the network can be geometry-aware and preserve the geometric information of the target meshes (i.e., the shape). Due to the page limitation, and as SPAdaIN is not the contribution we want to emphasize, we put the details of how we design the network (e.g., the choice of the SPAdaIN and attention structure) and the specific dimension of each layer and hyperparameters of the network are attached in the Appendix. As suggested, we will add some explanation of each introduced component in the paper in the new version.
>
> **Q2: The design of the cross-attention mechanism depicted in Fig 2 is not clearly described. More detailed explanations regarding its implementation are required.**
>
> A2: We appreciate your suggestion; the cross-attention block in Fig. 2 is originally demonstrated in the paper with Eq. (1), where q and k are the latent pose codes from the target mesh and driving pose. We will enhance the illustration of Fig. 2 and indicate that the cross-attention block (green on) is based on Equ. (1) with better indications in the new version.
>
> **Q3: The proposed encoder is named "geometry adaptive 3D feature encoder," but it seems that geometry is only involved in the positional embedding. It would be beneficial to consider incorporating a geometry-aware design in the encoder in addition to the positional embedding.**
>
> A3: Thanks for your suggestion. As discussed in Q1. The geometry-aware function is achieved via SPAdaIN in the decoder part of LART, where it takes the raw target mesh as residual inputs to learn the spatial geometry, which is more efficient for the generation. Meanwhile, our encoder focuses more on the pose presentations and the pose correspondence learning between the target mesh and source pose. If we also introduce spatial and geometric aware design in the encoder, the network might be redundant and intricate as the complicated and too detailed geometric (such as wrinkles and tissues) are less necessary for the pose encoding.
>
> **Q4: The paper lacks an ablation study on the three different positional embedding methods presented in Fig 3. It is important to analyze the performance of each positional embedding method, as the supplementary Table 7 only reports the loss without providing any analysis.**
>
> A4: We appreciate that reviewer notice the ablation study of the embedding methods. Firstly, compared to concatenating embedding, adding embedding takes fewer memory allocations and is proven to perform better than (please see ViT [A Dosovitskiy · 2020], PoseFormer [C Zheng · 2021]), thus in this work, we take adding embedding by default. As the reviewer noticed, Table 7 is the ablation study of different embedding methods. However, we find it tricky to conduct a fair comparison with a good protocol. Precisely, our adaptive embedding is proposed specifically for target mesh with different vertex sizes than the motion sequences, while the fixed scheme embedding cannot; thus, we cannot directly compare those two schemes on different target sizes. As a result, we only report the evaluated reconstruction loss (i.e., the PMD used for evaluation) in training, in which the target meshes are all fixed size. This loss is equivalent to the evaluation of unseen motion settings which we use as the ablation study of different embedding schemes. Note that this loss is not fair enough as it only evaluates the model with target meshes with fixed vertex sizes which cannot fully demonstrate the strength of the adaptive embedding. We will clarify this in the paper.
>
> **Q5: While the paper includes visualizations in figures, it would be advantageous to provide a demo video to showcase the actual quality of the visualizations.**
>
> A5: Please see our response to the first concern of reviewer 1. In short, we include additional qualitative videos to showcase the more generating performance of our LART in folder ‘more results’, and provide experimental results with 4D raw scans as input to drive the target meshes in folder ‘4D raw scan input’ as well as noisy driving source in folder ‘noisy driving motion’. Furthermore, we demonstrate that our method can conduct linear operations to the latent space and achieve meaningful manipulation of the motions in folder ‘interpolation’.
>
> The result videos can be found at this anonymous link: https://we.tl/t-WDuWeIFY0K
>
> Please let us know if you have more questions or concerns; thanks!

---

> > ### Comment · Reviewer_cJpb · 2023-08-18
> >
> > Thanks for the response. I hope the author can revise the paper based on these suggestions in the updated version.

---

> > > ### Author Response · Authors · 2023-08-21
> > >
> > > We will revise the current version and enhance the manuscript based on your suggestion. We wish to express our appreciation for your constructive comments and corrections, which have greatly improved the manuscript.

---

### Official Review · Reviewer_SFNt · 2023-07-03

**Soundness:** 3 good
**Presentation:** 3 good
**Contribution:** 3 good
**Rating:** 5
**Confidence:** 4

**Summary:**

The paper presents a method to transfer motion from a dynamic input sequence to a static 3D object. There are several novel components presented in the method: a novel feature encoder with an adaptive positional encoding scheme and a novel latent geometric regularization on the transformer. The paper is evaluated using motions from the AMASS dataset and shapes from the DFAUST dataset.

**Strengths:**

The paper is the first to address the problem of retargeting motion from a motion sequence to a novel shape. In this regard, it is quite a novel paper. There are some novel components in the method as well as listed above. I like that the method can also work when the driving motion meshes are noisy.

**Weaknesses:**

The memory requirements of the method is a weakness; this is listed in the main paper itself.

**Questions:**

none

**Limitations:**

The authors have mentioned several limitations of their method in the main paper. I have mentioned a few of them above as well. But  despite the limitations and weaknesses, I think there is merit in the paper and it deserves acceptance. So I recommend a boderline accept rating.

---

> ### Author Rebuttal · Authors · 2023-08-10
>
> Thank you for your acknowledgment of the novelty of our work and the constructive feedback!
>
> **Q1: The memory requirements of the method are a weakness; this is listed in the main paper itself.**
>
> A1: Regarding the concern on the memory requirements, this memory allocation is predominantly attributed to the necessity of incorporating multiple frames as input data. The rationale behind this allocation is to empower the model to capture and comprehend temporal patterns with Latent Metric Regularization effectively. This, in turn, enables the model to capture and synthesize reasonable temporal dynamics (with locally linear in Euclidean space to gain better deformation effects). As discussed in the paper Section 3.1, few efforts have been made for end-to-end 3D motion transfer due to the substantial computational consumption. To our knowledge, it is the first attempt to achieve end-to-end 3D motion transfer with a customized Transformer architecture.
>
> **Q2: The authors have mentioned several limitations of their method in the main paper. I have mentioned a few of them above as well. But despite the limitations and weaknesses, I think there is merit in the paper, and it deserves acceptance. So I recommend a borderline accept rating.**
>
> A2:
> We sincerely appreciate your acknowledgment of the merit in our work and your recommendation for a borderline accept rating and we are pleased that you recognize the significance of our work despite the identified limitations.
>
> Please let us know if you have more questions or concerns; thanks!

---

> > ### Comment · Reviewer_SFNt · 2023-08-21
> >
> > Thanks for taking the time to reply to my review. I maintain that this paper has merit and I will retain my initial rating for the paper.

---

> > > ### Author Response · Authors · 2023-08-21
> > >
> > > Thank you for replying to our response and we wish to express our appreciation for your positive comments!

---

### Official Review · Reviewer_gKzm · 2023-07-05

**Soundness:** 3 good
**Presentation:** 3 good
**Contribution:** 2 fair
**Rating:** 5
**Confidence:** 5

**Summary:**

This paper describes a method to transfer the dynamic mesh sequences to the unseen 3D mesh target. A transformer-based model is developed to implicitly learn the correspondence. In this model, pose and identity embeddings are separately encoded from the meshes. A decoder is designed to generate mesh sequences while considering the target mesh and encoded embeddings. The proposed method, LART, has been mainly evaluated on DFAUST dataset.

**Strengths:**

1. Transferring 3D mesh motion to the 3D unseen target is very challenging in learning the correspondce between the input and the targetm esh. In this paper, a transformer-based model is trained to implicitly learn such correspondence. The idea is interesting.
2. The paper is well written to clearly state the differences compared with previous methods and the main contributions.

**Weaknesses:**

The main concern is about the experiments.
The task is not claimed as transferring 3D motions between general subjects, not only for human meshes. While the evaluation, especially quantitative evaluation, is only about human. There are only a few qualitative results shown in the Fig. 5 to demonstrate the generalization ability of the proposed method, which may not be convincing enough. As a temporal method, it would be better if a video could be provided to show its visual performance. Therefore, the contributions may be a little over-claimed. Current experiments may not be enough to support them.

**Questions:**

I think the most attractive point of the proposed method is its generalization ability.
Is there any other way to quantitatively / qualitatively demonstrate the generalization ability of the proposed method?

**Limitations:**

Yes.

---

> ### Author Rebuttal · Authors · 2023-08-10
>
> Thank you for your acknowledgment of the challenge of our work and the constructive feedback! We will address your questions and concerns in the following:
>
> **Q: The main concern is about the experiments. I think the most attractive point of the proposed method is its generalization ability. Is there any other way to quantitatively / qualitatively demonstrate the generalization ability of the proposed method?**
>
> A: Please see our response to the first concern of reviewer 1. In short, we include additional qualitative videos to showcase the more generating performance of our LART in folder ‘more results’, including various motions, and providing experimental results with 4D raw scans as input to drive the target meshes in folder ‘4D raw scan input’, as well as noisy driving source in folder ‘noisy driving motion’. Furthermore, we demonstrate that our method can conduct linear operations to the latent space and achieve meaningful manipulation of the motions in the folder ‘blending and interpolation’. All those abilities demonstrate the generalization ability of our LART.
>
> The result videos can be found at this anonymous link: https://we.tl/t-WDuWeIFY0K
>
> Please let us know if you have more questions or concerns, thanks!

---

> > ### Author Response · Authors · 2023-08-18
> > **Further discussion**
> >
> > Dear Reviewer gKzm,
> >
> > Thank you so much again for your time and efforts in assessing our paper. Hope our rebuttal has addressed your concerns. We are happy to discuss with you further if you still have other concerns before the rebuttal due.
> >
> > Thanks for helping improve our paper.

---

> > > ### Comment · Reviewer_gKzm · 2023-08-19
> > > **Thanks for all these human mesh videos**
> > >
> > > These results have largely resolved my concerns about generalization in transferring human motion sequence. Therefore, I have upgraded my rating.
> > >
> > > However, the claimed contributions are transferring motion of general objects, like hand or elephant, not just human motion. So we still lack more convincing evidence for more general objects. Therefore, the paper might still suffer from the overclaimed contribution in generalization ability. Please consider to properly state the contribution.

---

> > > > ### Author Response · Authors · 2023-08-21
> > > >
> > > > We appreciate your reply!
> > > >
> > > > Since the rebuttal due is ending soon when we receive your latest comment, we cannot provide the animals and hands video results on time. But we will prepare a video link demonstrating the video results on animals and hands in one or two days.

---

### Official Review · Reviewer_hZ8Y · 2023-07-07

**Soundness:** 4 excellent
**Presentation:** 2 fair
**Contribution:** 4 excellent
**Rating:** 6
**Confidence:** 3

**Summary:**

The paper proposes to improve the SOTA of the learned pose/motion transfer on unrigged 3D meshes. The architecture consists of a geometry adaptive feature encoder, a LART decoder, and a latent metric regularizer.

The geometry adaptive feature encoder first extracts features similar to NPT [33] by casting each vertex to a higher dimensional feature, e.g. 1,024 dimensions, through a series of 1D convolutions followed by an instance norm. A typical approach then is to either concatenate or add the feature to the embedding, which limits the model to only work on a fixed number of vertices. Instead, the paper proposed to apply max pooling across all vertices of the extracted feature, add to the embedding, then tile this to an arbitrary number of vertices.

The resulting feature is passed to the LART decoder which is essentially a transformer to attend to the corresponding vertex in the driving mesh and the target mesh. This is unlike 3D-CoreNet [29] where the correspondence is learned explicitly with the optimal transport.

Finally, the model applies the latent metric regularization to encourage the poses in motion to be interpolatable.

**Strengths:**

All proposed components are novel and reasonable to achieve the goal. In particular, the use of the transformer to implicitly learn the 3D geometry correspondence is a great idea.

**Weaknesses:**

No videos are provided, making it hard to qualitatively discuss the method.

**Questions:**

Can authors provide videos of various results, as the paper's title is "motion transfer?" How well does the method handle temporal coherency? Can authors share the video of the ablation on the latent metric regularization?

Have authors tried the method on a driving sequence with varying vertex count, e.g. raw 4D scan output? In theory, the method seems to be able to handle this.

Can authors demonstrate the effect of the flattening by applying algebraic operation? Would it make sense to show the results of motion blending and motion in-betweening?

Is there a way to evaluate correspondence learning compared to others like DiffusionNet [Attaiki et al. 2022]?

**Limitations:**

As the authors infer, the method will not handle geometric and physical constraints like volume preservation and collisions. I understand that such effects are not in the scope of this work.

---

> ### Author Rebuttal · Authors · 2023-08-10
>
> Thank you for your acknowledgment of the novelty of our work and the constructive feedback! We address your questions and concerns in the following.
>
> **Q1: Can authors provide videos of various results, as the paper's title is "motion transfer?" How well does the method handle temporal coherency? Can authors share the video of the ablation on the latent metric regularization?**
>
> A1:  The result videos can be found at this anonymous link:
> https://we.tl/t-WDuWeIFY0K
>
> In the folder ‘SOTA comparison’, we visualize the results of both our model and other STOA methods (3DCoreNet, NPT, LART w/o LMR, LART, and Ground Truth). We summarize the advantages of our LART over others in the following:
>
> (1) Comparison with SOTA methods: with the qualitative results shown in the video, our method performs better than existing SOTA methods in the visual effects, while the quantitative evaluation in the paper also proves it.
>
> (2) Temporal coherency with LMR ablation study: as one can see, although the LART (both with and without LMR) outperforms other SOTA methods in general, the LART without LMR still has observable artifacts such as shrinking arms and body while the LMR can substantially improve the temporal coherence.
>
> **Q2: Have authors tried the method on a driving sequence with varying vertex count, e.g., raw 4D scan output? In theory, the method seems to be able to handle this.**
>
> A2: Thanks for your suggestion. We followed your suggestion and added additional experiments as requested:
>
> Setting: We use the official DFAUST release raw 4D scans, with subject 50027, motion ‘shaking arms’ as input.
> We randomly sampled 6,890 points from the raw scans (since the vertex point number of the raw scan is too huge with more than 15,000 points) and conducted the motion transfer with a pre-trained LART to verify its performance both with and without any further finetuning settings. The result videos can be found at the above anonymous link, folder ‘raw scan input’.
>
> As we can see, using a pre-trained model without any not finetuning to transfer motion from raw 4D scans is extremely challenging (existing methods all need domain-specific finetuning to take raw data as inputs, such as MetaAvatar [S Wang 2021, NeurIPS]). Because the domain distribution of watertight meshes in the training and raw scans in the testing are quite different. For instance, the distortion on the hands in the video is caused by the too-sparse point sampling on the hands with the naive random sampling, while the points on watertight meshes are evenly given. Thus, we further finetuning the model on 4D, the raw 4D scan for ten epochs, the visual results get much better, and the domain de. To conclude, the demo shows the potential of our LART to work directly on 4D raw scans with domain-specific finetuning, which could be the future direction.
>
> **Q3: Can authors demonstrate the effect of the flattening by applying algebraic operation? Would it make sense to show the results of motion blending and motion in-between?**
>
> A3: Thanks for your valuable suggestion. Exploring the potential of applying algebraic operations directly on the latent space would be interesting. We demonstrate two examples by 1) taking two frames from the same motion, i.e., interpolating (boxing), and 2) taking two frames from different motions (one_leg_loos and chiken_wings), i.e., blending, as the first and last frame to generate the corresponding mid-frame outputs by varying the alpha in Equ (3). We attach the results in the ‘blending and interpolation’ folder; as we can see, our LART has the potential to generate reasonable ‘in-between’ outcomes: the intermedia pose has both lifting leg and raising arm actions.
>
> **Q4: Is there a way to evaluate correspondence learning compared to others like DiffusionNet [Attaiki et al. 2022]?**
>
> A4: We really appreciate your valuable suggestion to find a way to evaluate correspondence learning and mention the DiffusionNet [Attaiki et al. 2022]. DiffusionNet focuses on the 3D shape-matching problem. Although it’s similar to 3D motion transfer to some extent, it belongs to another research and beyond our research scope. Although aligning the correspondence can be beneficial to the motion/pose transfer, it’s not the ultimate goal in our 3D motion transfer task. One can still achieve robust motion/pose transfer without the need for correspondence alignment. Thus, we think quantitatively evaluating the correspondence learning with rigorous metric evaluation might go too far from the original task and might mislead the research direction. However, we agree with the reviewer that it would be meaningful to quantitatively evaluate and verify the correspondence learning with the LART in different tasks, such as shape matching and deformation, as it shows promising qualitative results, but we will put it as future work.
>
> **Q5: As the authors infer, the method will not handle geometric and physical constraints like volume preservation and collisions. I understand that such effects are not in the scope of this work.**
>
> A5:
> We agree with your comment that handling geometric and physical constraints can effectively improve the visual effect. How to effectively introduce those physical prior into the learning is a promising direction for the 3D motion transfer.
>
> Please let us know if you have more questions or concerns; thanks!

---

> > ### Comment · Reviewer_hZ8Y · 2023-08-20
> >
> > As the reviewer gKzm says, I wish authors provided video results on animals and hands.
> >
> > I still wish the paper compares against other correspondence learning techniques since the paper is titled "Neural Correspondence Learning." I disagree with the authors' comment "one can still achieve robust motion/pose transfer without the need for correspondence alignment." The pose transfer problem boils down to the correspondence problem. I suggest authors to change the paper title if there will be no evaluations in terms of the correspondence.
> >
> > I also suggest authors cite Neural Jacobian Fields [Aigerman et al. 2022] and consider comparisons and discussions.
> >
> > For now, I will keep my score but I feel the paper is weaker than my initial review.

---

> > > ### Author Response · Authors · 2023-08-21
> > >
> > > Many thanks to your reply!
> > >
> > > Since the rebuttal due is ending soon when we receive your latest comment, we cannot provide the animals and hands video results on time. But we will prepare a video link demonstrating the video results on animal and hands in one or two days.
> > >
> > > Regarding correspondence learning, we will conduct a preliminary quantitative evaluation by comparing the reconstruction error of our work with other methods to showcase correspondence learning ability. The experimental results will be provided in one or two days.
> > >
> > > We appreciate the reviewer mentioning Neural Jacobian Fields, the work we were also interested in. Intead of using generative networks like ours, the Neural Jacobian Fields uses mesh-specific, basic linear-algebra operations to an intrinsic field of matrices defined over the tangent spaces to preserve the detailed geometry. It's very computational efficient and effective. However, it relies heavily on discrete
> > > differential geometry operators, which needs rigorous processed watertight meshes, while our LART use simple LMR to constrain the deformation which is more capable to complex inputs such as 4D raw scans. We will add discussions about Neural Jacobian Fields in the revised version.

---

### Author Rebuttal · Authors · 2023-08-10

We would like to thank all the reviewers and AC for dedicating your time and expertise to assess our manuscript thoroughly. We are glad by the positive remarks from the reviewers on various aspects of our work (the novelty of LART, the versatility of the proposed method, the robustness of handling noisy input, the generalization ability, and the paper being well written).

We have taken each reviewer's suggestions and comments into consideration and have made comprehensive replies in the rebuttal, including a further explanation of specific technical details, addressing some ambiguous issues, and as well as various visualized experimental results requested by the reviewers.

We have provided individualized responses to each reviewer's comments in the box below. The extra experimental results are attached in the anonymous link:
https://we.tl/t-WDuWeIFY0K

---

> ### Author Response · Authors · 2023-08-17
>
> Dear reviewers,
>
> Many thanks again for your constructive comments.
>
> You are welcome to provide us feedback if any as the open discussion phase ends soon.
>
> We updated the link to the video demo as the previous one expired in case some reviewers still didn't download it:
>
> https://we.tl/t-n8pzoU0LGV
>
> We are glad to answer any follow-up questions.
>
> Many thanks,
>
> Authors

---

### Decision · Program_Chairs · 2023-09-21

**Decision:**

Accept (poster)

**Comment:**

The paper introduces a technique for transferring motion from a dynamic mesh sequence to a previously unseen 3D mesh target. Initially, the paper received borderline to positive scores. The rebuttal effectively addressed several concerns raised by the reviewers, resulting in positive evaluations from all reviewers in the end. The AC agreed with the reviewers' assessments and made the decision to accept the paper.

The authors are advised to moderate their claim regarding correspondence learning and adjust the paper title accordingly. Additionally, they are strongly encouraged to include supplementary material containing additional experiments on correspondence learning, as well as results pertaining to animals and hands, as previously promised.